# RhoGTPase in Vascular Disease

**DOI:** 10.3390/cells8060551

**Published:** 2019-06-06

**Authors:** Derek Strassheim, Evgenia Gerasimovskaya, David Irwin, Edward C. Dempsey, Kurt Stenmark, Vijaya Karoor

**Affiliations:** 1Cardiovascular and Pulmonary Research Lab, Department of Medicine, Anschutz Medical Campus, University of Colorado, Aurora, CO 80045, USA; derek.strassheim@ucdenver.edu (D.S.); Evgenia.gerasimovskaya@ucdenver.edu (E.G.); David.irwin@ucdenver.edu (D.I.); edward.dempsey@ucdenver.edu (E.C.D.); kurt.stenmark@ucdenver.edu (K.S.); 2Department of Pediatrics, Anschutz Medical Campus, University of Colorado, Aurora, CO 80045, USA; 3Pulmonary Sciences and Critical Care Medicine, Department of Medicine, Anschutz Medical Campus, University of Colorado, Aurora, CO 80045, USA; 4Rocky Mountain Regional VA Medical Center, Aurora, CO 80045, USA

**Keywords:** RhoGTPases, endothelial, smooth muscle, fibroblasts, vascular, therapeutics

## Abstract

Ras-homologous (Rho)A/Rho-kinase pathway plays an essential role in many cellular functions, including contraction, motility, proliferation, and apoptosis, inflammation, and its excessive activity induces oxidative stress and promotes the development of cardiovascular diseases. Given its role in many physiological and pathological functions, targeting can result in adverse effects and limit its use for therapy. In this review, we have summarized the role of RhoGTPases with an emphasis on RhoA in vascular disease and its impact on endothelial, smooth muscle, and heart and lung fibroblasts. It is clear from the various studies that understanding the regulation of RhoGTPases and their regulators in physiology and pathological conditions is required for effective targeting of Rho.

## 1. Introduction

Ras homologous (Rho) GTPases are a family of 20 proteins belonging to the small GTPases link surface receptors to the organization of the actin cytoskeleton and are essential in many cellular processes [1]. In the vasculature, Rho signaling pathways are involved in the regulation of endothelial barrier function, inflammation and trans-endothelial leukocyte migration, platelet activation, thrombosis, and oxidative stress, as well as smooth muscle contraction, migration, proliferation and differentiation, and adventitial fibrosis [2,3,4,5]. Rho GTPases are active in the GTP-bound form and inactive when bound to GDP. Rho-GDP is sequestered in the cytoplasm by binding to Rho guanine dissociation inhibitors (Rho GDIs) leading to inactivation. Rho guanine nucleotide exchange factors (RhoGEFs) catalyze the exchange of GDP for GTP to activate RhoA. Activation is turned off by GTPase-activating proteins (RhoGAPs) that induce the hydrolysis of GTP to GDP (Figure 1) [6]. Rho GTPases are regulated by post-translational modifications that include isoprenylation, carboxymethylation, oxidation, nitration, phosphorylation, and ubiquitination [7,8,9]. Isoprenylation of the C-terminus of Rho GTPases enhances their binding to the cell membrane, a characteristic that is important for interaction with signaling effectors, which include Rho-associated kinase (ROCK) 1, ROCK2, mammalian diaphanous (mDia), Rhophilin-Rhotekin, Citron, and protein-kinase N (Figure 1) [5]. RhoA phosphorylation of Ser188 by cyclic AMP-dependent protein kinase (PKA) and cGMP-dependent protein kinase (PKG) causes its localization in the cytosol and inhibits the RhoA-Rho kinase pathway, thereby contributing to the vasodilator effect [7].

Most of the current knowledge of the Rho subgroup in disease pathogenesis been gained through experiments on Rho-subfamily, which include Rho, Rac, and Cdc42 and the effector ROCK. RhoA/Rho-kinase activation has significant effects on various cardiovascular diseases, mainly arterial hypertension, atherosclerosis, heart attack, stroke, and others, such as coronary vasospasm, venous diseases, myocardial hypertrophy, myocardial ischemia-reperfusion injury, and vascular remodeling [10,11]. Activation of Rho GTPase/ROCK pathway has been reported in various cardiovascular diseases (CVDs) [12], such as pulmonary arterial hypertension (PAH), chronic pulmonary obstructive disease (COPD) [13,14], idiopathic pulmonary fibrosis (IPF) [15], asthma [16], acute lung injury (ALI) [17], acute respiratory distress syndrome (ARDS) [3], cardiac hypertrophy [18], atherosclerosis [19], or restenosis [20].

Rho/Rho-kinase plays a crucial role in the development of cardiovascular disease by promoting endothelial cell (EC) barrier dysfunction [5], reactive oxygen species (ROS) production [21], and inflammation [22] in EC, contraction, migration, and proliferation [23,24] in vascular smooth muscle cells (VSMC), and transformation of fibroblasts to myofibroblast, proliferation of fibroblasts, and extracellular matrix synthesis in fibroblasts (Figure 2) [25]. In this review, we summarize the role of RhoA family proteins in vascular cells focusing on the diseases that involve endothelial barrier dysfunction inflammation, smooth muscle contraction, proliferation, and adventitial fibroblasts oxidative stress, in lung and heart.

## 2. Endothelial Cells

### 2.1. Role of RhoGTPase in Endothelial Barrier Function

Endothelial cells (ECs) form the inner lining of blood vessels and are important in maintaining vascular tone, leukocyte transmigration, thrombosis, angiogenesis, and immunity [27]. They form a barrier between blood plasma and tissues by regulation of intercellular junctions and controlling vascular homeostasis [28]. The activity of Ras-related C3 botulin toxin substrate (Rac) 1, cell division cycle 42 (Cdc42), and Ras-related protein (Rap) 1 is essential for the maintenance of microvascular endothelial barrier function under physiological conditions [29]. RhoA by increasing EC-contraction and disruption of EC-EC junctions is involved in promoting leak [30]. RhoA/ROCK promotes the dissociation of cell-cell adherens junctions (AJs) by suppressing Vascular endothelial (VE)-cadherin expression, its membrane localization, and expression of occludin and claudin-1, the major components of tight junctions [30,31].

The vascular leak occurs in various pathological conditions, including ventilator-induced acute lung injury, sepsis, asthma, PAH, and COPD [2]. In sepsis, the decrease in endothelial barrier function by lipopolysaccharide (LPS) and tumor necrosis factor (TNF) α increases microvascular permeability and contributes to multi-organ failure and death [2,30,32]. Cytokines, such as interleukin (IL)-1, TNFα, in PAH and other CVDs can activate RhoA promoting EC contraction and weakening of adherens junctions [33]. Inflammatory stimuli increase ROS levels and the opening of inter-endothelial junctions and promote the migration of inflammatory cells across the endothelial barrier [34]. G protein-coupled receptors (GPCR), such as angiotensin (Ang)II, endothelin (ET)1, thrombin, and other vasoconstrictors, are elevated in pulmonary arterial hypertension (PAH) promoted vascular leak, by activating RhoA [35]. Activation of the transcription factor Yes-associated protein (YAP) leads to endothelial dysfunction, where the EC junction protein junctional cadherin 5 associated (JCAD) activates YAP pathway, by relieving the tonic inhibitory effects of kinase linker for activation of T cells ( LATs) on YAP, an effect which requires RhoA [36]. In diabetes, RhoA and Receptor for advanced glycation endproducts (RAGE) form a complex called RhoA/RAGE, to induce Rho-kinase activation, resulting in the reorganization of the actin cytoskeleton, leading to endothelial cell hyperpermeability [37]. In the lungs, hypoxia increases vascular permeability by activation of RhoA-Rac1 [38,39,40,41,42,43]. PI3K, RhoA, and ROCK1 are involved in hypoxia-regulated ATP exocytosis and ATP-induced angiogenic responses in vasa vasorum endothelial cells, suggesting that these pathways constitute an autocrine/paracrine loop of ATP release and signaling and contribute to pathological vascular remodeling [44]. Cigarette smoke disrupts the endothelial barrier in the lung by increasing ROS, leading to activation of RhoA and endothelial dysfunction [45].

Increase in cAMP by adenosine is barrier protective and inhibits RhoA activation in a sepsis model of lung injury [46]. Barrier protection by prostacyclin PGI2 is mediated by activation of Rap1-EPAC (Exchange protein directly activated by cAMP), and vascular leakage was found to be worse in Rap1a(-/-) mice [29]. PGI2 also protects against thrombin or mechanical stretch-induced vascular permeability by increasing the krev interaction trapped-1 (KRIT1) activity in a Rap1 dependent manner [47]. Adaptor protein (KRIT1) stabilizes EC adherens junctions, increasing KRIT1 VE-cadherin binding adherens junctions, leading to EC barrier enhancement. Activation of Rac signaling inhibits Rho activity, via p21 activated kinase (PAK) 1-dependent inhibition of p115RhoGEF activity, attenuates Rho-mediated barrier disruption and may contribute to the maintenance of EC monolayer integrity in injured lungs [48].

The beneficial effect of statins in vascular leak protection is due to inhibition of the geranyl-geranylation enzyme required for RhoA post-translational modification and activity [49,50]. Atorvastatin reduces vascular leak in diabetic mice by down-regulation of RhoA and up-regulation of Akt/GSK3 [51]. HDAC inhibitor studies suggest protection against sepsis-induced vascular leakage by suppressing Hsp90-dependent RhoA activity and signaling [52]. However, studies using RhoA inhibitor revealed that basal RhoA activity was also essential to maintain barrier integrity by regulating VE-cadherin levels [53].

### 2.2. RhoGTPase in Venous Endothelial Dysfunction

Several studies have shown a direct relationship between venous endothelial function and cardiovascular events, such as pulmonary embolism, venous thromboembolism, and venous insufficiency [54]. Risk factors for these events include older age, smoking, obesity, diabetes, atherosclerosis, and pulmonary disease [55,56]. Rho kinase pathway modulates endothelial fibrinolytic activity by up-regulating thrombogenic molecules like platelet-activating factor plasminogen activator inhibitor-1 [57] and tissue factor [58] and fibrogenic molecules like transforming growth factor-β1 (TGFβ1) [59]. Rac1 inhibition improves venous endothelial function and reduces NADPH oxidase activity in saphenous vein grafts harvested from patients with vascular diseases undergoing peripheral bypass surgery [60]. The long-term Rho-kinase inhibition by fasudil significantly suppresses intimal hyperplasia rabbit vein grafts after the bypass surgery [61]. In contrast, Rho kinase activity is lower in varicose veins and associated with decreased levels of fibronectin expression, leading to reduced contractility of smooth muscle cells [62].

### 2.3. Role of RhoGTPase in Vascular Inflammation

Rho family of small GTPases and their immediate downstream effectors are important mediators of vascular inflammation. Distinct Rho proteins are involved in the positive or negative regulation of Nuclear factor kappa B subunit (NFκB) in different settings. RhoA and RhoB can have either a positive or negative regulatory role depending on the context. Rac1 and cdc42 are activated for Nuclear factor kappa B subunit (NFκB) activity. RhoA-induced NFκB activity is also significantly suppressed by the expression of RhoH, a GTPase deficient member of the Rho family. [63]. RhoA and RhoB can have either a positive or negative regulatory role depending on the context [64].

Activation of RhoA/ROCK is required for adherence and trans-endothelial migration of monocytes and neutrophils [65,66,67], and in production of inflammatory mediators by vascular smooth muscle cells (SMCs), cardiomyocytes [68,69], monocytes, and T-cell activation [70]. Gene-targeting studies using myeloid lineage-specific drivers showed that RhoGTPases were involved in various steps of polarization, integrin-mediated spreading, actin polymerization, chemotaxis, and phagocytosis of immune cells [71,72]. Rac members of the Rho family are important in innate immune cells functions, and their deficiency attenuated migration in vivo [73,74]. Rac2 is essential for primary granule release and NADPH oxidase activation, both prerequisites for microbial killing [75]. RhoA maintains neutrophil quiescence and is required for suppressing hyper-responsiveness. Rho-deficient hyper-activated neutrophils exaggerate tissue injury in LPS-induced ALI [76]. Vascular injury initiates rapid platelet activation that is critical for hemostasis, but it also may cause thrombotic diseases, such as myocardial infarction and ischemic stroke [77]. RhoA is an important regulator of platelet function in thrombosis and hemostasis [78]. RhoA deficiency in platelets increases tail bleeding times and is protective in different models of arterial thrombosis and a model of ischemic stroke [78]. In a carotid artery ligation model using wild type and heterozygous ROCK1+/- mice, leukocyte-derived Rock1 was shown to contribute to neointima formation and leukocyte infiltration [79]. Experiments with reciprocal bone marrow transplantation revealed that WT to Rock1+/– transplantation resulted in increased neointima formation [79]. Evidence, from inhibitor studies, suggests that RhoA/Rho kinase signaling in vascular and hematopoietic cells participate in multiple steps of atherogenesis, including leukocyte recruitment, cytokine, and chemokine release; and oxidized Low-Density Lipoproteins (LDL) uptake in macrophage and foam cell formation [33]. In atherosclerosis, macrophage deletion of G_12/13_ reduces inflammatory potential and RhoA activation [80]. Deletion of RhoA down-regulates monocyte/macrophage CX3CR1/CX3CL1 signaling inhibits macrophage infiltration and vessel occlusion and abrogates chronic rejection of mouse cardiac allografts [81].

Aortic aneurysm involves chronic inflammation of the vessel wall, VSMC senescence, oxidative stress, increased local production of proinflammatory cytokines, and increased activities of Matrix MetalloPeptidase (MMPs) [82]. Chronic Ang II infusion into apolipoprotein E-deficient mice promotes aortic aneurysm formation by Rock1 activation and cyclophilin secretion [83]. Fasudil, the Rho-kinase inhibitor, has reduced Ang II-induced aortic aneurysm formation [84,85,86]. The activity and the expression of Rock1 are enhanced at the inflammatory/arteriosclerotic coronary lesions [12].  Long-term treatment with the stress hormone cortisol causes coronary hyper-reactivity through the activation of Rho-kinase in pigs in vivo [87].

RhoA/Rock signaling is a significant player in the pathogenesis of systemic arterial inflammation, neointimal formation, and arteriopathy of PAH [88]. RhoA is shown to mediate phosphorylation of IkB and cause translocation of the dimers to the nucleus, although the mechanism is not known [64]. There is evidence that Rac1 regulates cytokine-stimulated NFkB activation via a redox-dependent pathway [89]. Expression of constitutive active RacV12 resulted in a significant increase in intracellular ROS, whereas dominant-negative Rac markedly reduced the increase in ROS as well as the activation of NFkB after cytokine stimulation [89].

In COPD, leukocyte activation by cigarette smoke extract (CSE) occurs in a RhoA-dependent manner, increasing endothelial CXCL16-leukocyte CXCR6 adhesion via NADPH oxidase (Nox)5 expression and RhoA/p38 mitogen-activated protein kinase (MAPK) /NF-κB activation [45]. The movement and secretion of platelet granules are dependent on RhoA signaling [90]. Platelets from patients, with certain forms of PH, are hyper-activatable and exhibit enhanced RhoA activity [91]. RhoA knockdown in T cells inhibits both Stromal Derived Factor (SDF)-1α–mediated trans-endothelial migration and f-Met-Leu-Phe (fMLF)-induced chemotaxis [76]. T cells regulate vascular remodeling in PAH in a RhoA dependent manner [92]. Increased Th17/Treg ratio in PAH, with increased IL-17, promote VSMC proliferation, which is reduced by ROCK inhibitors [36].

Agents that increase cAMP cause inhibition of RhoA and have anti-inflammatory properties. The anti-inflammatory action of prostanoid like Epoxyeicosatrienoic acids (EETs) on several cell types, including leukocytes, involves RhoA inhibition [93]. Clopidogrel, prescribed to reduce strokes, heart disease, is an irreversible antagonist of purinergic receptor P2Y12R, inhibits adenylyl cyclase and increases cAMP-PKA- Exchange protein directly activated by cAMP (EPAC), reducing RhoA activation [94]. Impaired bacterial phagocytosis can be restored in neutrophils using the drug, 8CPT-2Me-cAMP, which activate EPAC, through Rap1 [39].

## 3. Smooth Muscle Cells

Rho/ROCK plays an essential role in vascular smooth muscle cell (VSMC) contraction, actin cytoskeleton organization, adhesion, and cytokinesis and cell migration, proliferation, apoptosis/survival, gene transcription, and differentiation [95]. RhoA and its target ROCK, mediate VSMC contraction, stress fiber formation, cell migration, and, indirectly, blood pressure regulation [96]. SMC proliferation and migration contribute to vascular remodeling in atherosclerosis, hypertension, and restenosis [97,98,99]. Activation of Rho and its effector ROCK, by GPCR agonists coupling to Gα_12/13_ proteins, leads to the phosphorylation of the myosin light chain of myosin II, through inhibition of myosin phosphatase and Ca^2+^ independent regulation of smooth muscle contraction [100]. The ROCK-mediated enhancement of smooth muscle contraction occurs in the absence of significant changes in [Ca^2+^] and is, therefore, considered a Ca^2+^ sensitization mechanism [101]. ROCK effects on actin remodeling are important in controlling cell migration [101]. Rho/ROCK also is involved in the expression of differentiation genes that define the contractile phenotype of smooth muscle [78,102]. RhoA-dependent regulation of the actin cytoskeleton selectively regulates SMC differentiation by modulating Serum response factor (SRF)/myocardin-dependent transcription of genes for contractile proteins [103]. ROCK activation is associated with increased vascular stiffness through processes that increase SRF/myocardin transcription [104,105]. Rac1 activity decreases the expression of contractile proteins in VSMCs when grown on fibronectin, suggesting that the balance of Rac and Rho activity and matrix determine SMC function [106]. RhoA signaling thus may serve as a convergence point for the multiple signaling pathways that regulate SMC differentiation.

Rho-kinase in VSMCs plays a crucial role in the pathogenesis of coronary artery spasm and contributes to angina, myocardial infarction, and sudden death [107,108]. Various preclinical models of hypertension have shown a role for Rho and its regulators in hypertension, which is predominantly due to enhancing contraction of SMCs [96]. Mice that are deficient for Gα_12/13_ or Rho guanine nucleotide exchange factor 12 (LARG), a GEF for Rho, are protected from salt-sensitive hypertension [109,110,111]. SMC-specific knockout of the related GEF, p115RhoGEF, inhibits the development of hypertension in response to AngII [112]. Arhgef1 plays a significant role in Ang II/AT1 receptor-induced RhoA activation in smooth muscle cells, vasoconstriction, and hypertension [113], and Arhgef1/RhoA signaling is turned on by renin-angiotensin-aldosterone-system (RAAS) activation in humans [114]. In contrast, mice deficient in Rho-specific GTPase Activating Protein ( GRAF) 3, an SMC specific RhoGAP, show increased basal and AngII-induced hypertension [115]. In patients with hypertension, an increase in Rho guanine nucleotide exchange factors (Rho-GEFs) and protein phosphate 1 (MYPT1) levels are observed [116]. Gene transfer of dominant-negative Rho-kinase reduces the neointimal formation of the coronary artery in pigs [117]. The protein caveolin (Cav) 1 suppresses RhoA activity in Pulmonary artery Smooth Muscle Cells (PASMCs), thereby protecting against vascular remodeling. The mechanism involves Cav1 binding to Gα13, preventing it from activating RhoA-GEF, p115RhoGEF [118,119,120,121,122]. Rock activation occurs in various models of both primary and secondary PAH [123]. Alterations in the activity of RhoA regulators, RhoGAP and RhoGDI, causes sustained activation of Rho and Rac and decreased contractile proteins expression in pulmonary artery SMCs [124]. A recent study has shown that Rho enhances PASMC proliferation by suppressing nuclear translocation of Smad1 by Bone Morphogenic Protein ( BMP) [125]. RhoA-Rock and ROS in obstructive sleep apnea syndrome induce chronic intermittent hypoxia, resulting in elevated blood pressure (BP) [126]. Smoking-induced increase in coronary artery spasm is due to increased RhoA activity [127].

## 4. Fibroblasts

Differentiation of fibroblasts into the smooth muscle-like myofibroblast is essential in wound healing and fibrosis [128]. Increase in adventitial myofibroblasts occurs in vascular diseases, including systemic [129] and pulmonary hypertension [130], vein graft remodeling [131], coronary transplant vasculopathy [132], and inflammatory abdominal aortic aneurysms [133]. Fibroblasts in the adventitia control homeostasis of the collagen matrix surrounding the vessels, protecting them from excessive stretch [134].  In response to injury, adventitial fibroblasts migrate and synthesize the extracellular matrix proteins and express α-smooth muscle actin. The excessive production of collagen I and growth factors are contributing to vascular remodeling, facilitating fibrosis, and attracting leukocytes to clear the damaged tissue and resolve fibrosis [135].

Transforming growth factor (TGF)-β1-Smad2 signaling is essential in the differentiation of cardiac [25], pulmonary [136], and aortic [137] adventitial fibroblast to myofibroblast. In rats with carotid artery balloon injury, perivascular treatment with dominant-negative N19RhoA attenuates neointimal formation and adventitial Smad2 phosphorylation [138]. Fibroblast deletion of Rock2 attenuates cardiac hypertrophy, fibrosis, and diastolic dysfunction [139]. Heterotrimeric Gi proteins link hedgehog signaling to activation of RhoGTPases to promote fibroblast migration [140].

Fibroblast proliferation in hypoxia-induced pulmonary hypertension requires activation of Rac1/p38 signaling cascade [141]. Inhibition of Rac1 or treatment with statins decreases adventitial fibroblast proliferation and p38 activation [4]. Rac1/ Signal Transducer and Activator of Transcription (STAT3) activation is involved in the proliferation and migration of fibroblast and medial hypertrophy in rats infused with AngII [142]. Pulmonary adventitial fibroblasts are highly sensitive to hypoxia and AngII, which cause fibroblast proliferation, the activation of fibroblast NAD(P)H oxidase, leading up to pulmonary hypertension [143].

### 4.1. Role of RhoGTPase in Vascular Oxidative Stress

Vascular ROS formation is stimulated by mechanical stretch, pressure, shear stress, hypoxia, and growth factors [144].  Changes in the vascular redox state are involved in the pathogenesis of atherosclerosis, aortic aneurysm, and vascular stenosis [145,146,147]. ROS increases RhoGTPase activity by oxidation of cysteine 18 in the p-loop of Rho GTPases, which results in GTPase activation independent of GEFs [21,148]. ROS increases the activity of low molecular weight protein tyrosine phosphatase (LMW-PTP), and subsequent elevation of p190 RhoGAP activity, resulting in indirect redox-dependent regulation of Rho GTPases [149]. Increased NO production by ECs leads to nitration of RhoA at Tyr34 and increases GTP exchange [150].

Endothelial cells generate ROS in response to diverse stimuli, including thrombin, histamine, TNF-α, inflammatory cytokines, LPS, glucose, cigarette smoke vasoactive peptides, cyclic stretch, or hypoxia, followed by reoxygenation [151,152]. ROS increases endothelial permeability and adhesion of leukocytes and their extravasation across the barrier [153]. Rho GTPase-dependent regulation of vascular permeability by Vascular Endothelial Growth Factor (VEGF), histamine, and inflammation require NO. Endothelial Nitric-oxide Synthase (eNOS)-/- mice and mice with mutation of the eNOS phosphorylation site S1176 show a decrease in vascular permeability and RhoA activation [154]. However, RhoA can regulate Reactive Nitrogen Species (RNS) production via negative regulation of eNOS expression suggesting complex interactions between Rho-kinase and NO signaling for endothelial homeostasis in vivo. Rho-kinase-deficient mice show preserved EC function in a diabetic model [155]. Rap1b(-/-) mice develop cardiac hypertrophy and hypertension due to defects in nitric oxide-dependent vasodilation [156]. Statins inhibit the synthesis of the isoprenoid intermediate required for post-translational modification of Rho and Ras needed for the activity. Statins up-regulate eNOS by cholesterol-independent mechanisms, involving the inhibition of Rho geranyl-geranylation [157]. Small GTP-binding protein dissociation stimulator (SmGDS) protein also plays a central role in the pleiotropic effects of statins, independently of the Rho-kinase pathway by decreasing Rac activation [157].

Rho-kinase is involved in ROS augmentation and neo-intima formation, in part, by promoting VSMC growth and stimulating pro-inflammatory signaling [158]. VSMC increased Nox-derived ROS generation enhances calcium signaling, up-regulates ROCK, and modulates the actin cytoskeleton, thereby improving vascular contraction and increasing vascular tone [159]. Oxidative stress affects calcium homeostasis by post-translational modification of calcium channels, Sarco/endoplasmic reticulum Ca(2+) ATPase (SERCA), and Transient receptor potential (TRP) channels, actin and actin-binding proteins, myosin, and cofilin [104]. Superoxide causes vasoconstriction while hydrogen peroxide induces both vasodilation and vasoconstriction, depending on the vascular bed and the Nox isoform involved [160]. Oxidative stress, endothelial dysfunction, and low-grade chronic inflammation contribute to the pathogenesis of coronary artery syndrome, leading to increased coronary smooth muscle Ca^2+^ sensitivity through RhoA/ROCK activation and resultant hypercontraction [127].

The vascular adventitia is an important site of vascular ROS production, and adventitial fibroblast NADPH oxidase-derived ROS is the sensor and messenger for the early development of vascular disease [161]. Rac1-deficient fibroblasts show decreased ROS levels and myofibroblast formation [162]. The local combination of adventitial fibroblasts with the activated leukocyte respiratory burst oxidase potentiates the production of adventitial ROS [143].

Intermittent hypoxia increases blood pressure in rats by increasing ROS and the RhoA-ROCK pathway [163]. RhoA-Rock and ROS in obstructive sleep apnea syndrome induce chronic intermittent hypoxia, resulting in elevated BP [126]. ROS production and Rho-kinase activation play a crucial role in myocardial damage after ischemia-reperfusion. Pretreatment with fasudil before reperfusion prevents endothelial dysfunction and reduces the extent of myocardial infarction in dogs in vivo [164]. The beneficial effect of fasudil has been demonstrated in a rabbit model of myocardial ischemia induced by an intravenous administration of endothelin-1 [165], a canine model of pacing-induced myocardial ischemia [166] and a rat model of vasopressin-induced chronic myocardial ischemia [167].

### 4.2. RhoGTPase in Heart and Lung

The beneficial effects of long-term inhibition of Rho-kinase for the treatment of cardiovascular disease are established in various animal models, such as coronary artery spasm, arteriosclerosis, restenosis, hypertension, multiple models of PH, stroke, and cardiac hypertrophy/heart failure [11]. Gene knockout studies have shown that Rho GTPase family proteins have an essential role in early mouse heart morphogenesis as they control a wide variety of cellular processes, such as cell morphology, motility, proliferation, differentiation, and apoptosis [168]. Global RhoA knock out (KO) is embryonic lethal [169]. Cardiac-specific overexpression of RhoA results in sinus and atrioventricular nodal dysfunction with significant dilatation of left ventricle and decrease in contractility in adult transgenic mice [170]. In another study, conditional knockout mice of RhoA in cardiomyocytes with MHC (myosin heavy chain)-Cre did not show an apparent abnormality in the heart under physiological conditions, but increased infarct size, and worsened the heart failure in myocardial ischemia/reperfusion(I/R) injury. Conversely, mice with cardiac-specific RhoA-overexpression are protected from I/R injury [171]. Cardiac-specific expression of constitutively active Rac1 has been shown to lead to either a lethal neonatal dilated cardiomyopathy or a resolving transient cardiac hypertrophy in [172]. Inhibition of Rho and Rac by cardiac expression of Rho GDIα disrupt cardiac looping and ventricular maturation and is embryonically lethal [168].

Rock is the downstream effector of Rho and has two isoforms, which are about 64% similar in sequence and have overlapping substrates. Rho-kinase enhances Ca^2+^ entry through activation of G-protein–coupled receptors, ventilatory hypoxia, and NOS inhibition [59]. In heart failure, Rock is involved in the regulation of myofibrillar Ca^+2^ sensitivity in cardiac muscle and contributes to irreversible myocardial damage [173,174]. ROCK is also involved in the pathogenesis of cardiovascular remodeling, and its inhibition plays a significant role in the treatment of the failing heart by limiting infarct size, which is the major contributor to the development of heart failure [175,176]. ROCK is activated in Type II diabetes, a condition which includes obesity, hypertension, various vasculopathies, and insulin resistance [177,178]. Studies with heterozygous and conditional KO ROCK1 and ROCK2 mice reveal different upstream activating/regulating mechanisms, different subcellular distributions of ROCK1 and ROCK2, and co-expression of different targets with ROCK1 and ROCK2 in different cell types [179,180]. In the heart, ROCK1 promotes fibrosis, while ROCK2 promotes hypertrophy [139,181]. Transgenic mice overexpressing an active form of ROCK1 in cardiomyocytes show spontaneous development of fibrotic cardiomyopathy, which is reversed by treatment with fasudil. Rnd3 haploinsufficient mice have increased ROCK activity and are predisposed to left ventricular pressure overload and develop severe heart failure after transverse aortic constriction with increased cleaved caspase-3 [182]. Rnd3 mice crossed with global ROCK1-knockout mice show significantly reduced cardiomyocyte apoptosis after the transverse aortic constriction. ARHGAP18 is a protective gene that maintains EC alignments in the direction of flow. Deletion of ARHGAP18 leads to the loss of EC ability to align and promotes atherosclerosis development [183].

ROCK activity is a biomarker of cardiovascular disease, and ROCK inhibition by statins occurs through cholesterol-independent mechanisms [157]. Mice with Rho-kinase inhibition in the developing heart (SM22α-restricted overexpression of dominant negative ROCK) spontaneously develop cardiac dilatation and dysfunction, myocardial fibrofatty changes, and ventricular arrhythmias, resulting in sudden death, a phenotype similar to Arrhythmogenic Right Ventricular Cardiomyopathy (ARVC) in humans with altered desmosome structure and Wnt signaling. However, mice with inhibition after birth using αMHC (*Myh6*)-directed expression of DN-ROCK did not develop the phenotype [170].

In the heart, studies have shown potential benefits of targeting Rho-ROCK, reducing cardiac hypertrophy, and fibrosis [176,184]. Intracoronary administration of fasudil and hydroxyfasudil inhibits coronary spasm in a porcine model [165]. Long-term treatment with a Rho-kinase inhibitor suppresses neointimal formation after vascular injury in vivo,  monocyte chemoattractant protein-1–induced vascular lesion formation constrictive remodeling, in-stent restenosis,  and development of cardiac allograft vasculopathy [10]. Perivascular fibrosis by myofibroblasts is considered a pathologic feature in cardiac hypertrophy [185,186].

Increasing evidence indicates a pathophysiological role for Rock in various animal models of pulmonary arterial hypertension and pulmonary fibrosis [88]. In idiopathic pulmonary fibrosis, fibroblasts show a decrease in Rnd3/p190RhoGAP pathway and an increase in RhoA activity [15]. Various ROCK inhibitors have shown benefit in multiple animal models of PAH with improvement in pulmonary hemodynamics, exercise capacity, and survival, summarized in the review by Zhang et al. [163]. Rho/ROCK activity is increased in bleomycin-induced fibrosis lung [187]. ROCKs also contribute to profibrotic cellular responses to lung injury, and ROCK inhibitors have shown to protect against damage [188,189]. The long-term effects of chronic treatment with fasudil include improving not only pulmonary hemodynamics and exercise capacity but also survival and have been demonstrated in animal studies.

### 4.3. Clinical Studies on the Role of Rho GTPase in Vascular Disease

ROCK activity is increased in patients with hypertension, pulmonary hypertension, stable angina pectoris, vasospastic angina, heart failure, peripheral artery disease, and stroke [10,58,190,191,192,193,194]. Enhanced activity of ROCK signaling is observed in the lungs of smokers [195]. Genetic analysis studies reveal the association of polymorphisms in RhoGTPase and regulators with vascular diseases. Polymorphisms in the ROCK2 gene are associated with a lower risk of developing hypertension [196]. Increased leukocyte ROCK activity is observed in patients with hypertension, pulmonary hypertension, metabolic syndrome, dyslipidemia, coronary artery disease, coronary vasospasm, left ventricular hypertrophy (LVH), and in heart failure with decreased systolic function [197]. Polymorphisms in Rock2 increase its activity and is associated with stiffer arteries [198]. In Caucasian women, single polymorphisms in ROCK1 show significant association with the risk of ischemic stroke [199]. In Korean individuals, Single Nucleotide Polymorphisms (SNPs) in Rock2 is protective in vasospastic angina. Individuals with specific SNPs in ROCK2 show increased leukocyte ROCK2 activity [200]. Two rare frequency variants of ROCK1 are thus found to significantly increase the risk of tetralogy de Fallot [201]. Genome-Wide Association Study (GWAS) studies have identified polymorphisms in FAM13A, a RhoGAP associated with COPD, in Chinese Han population [202].

Clinical studies with small populations of patients have demonstrated the beneficial effects of ROCK inhibitors [176]. Fasudil treatment prevents vasospasm associated with subarachnoid hemorrhage, acute ischemic stroke, angina, coronary artery spasm, atherosclerosis, and in the regulation of vascular tone in hypertensive renal transplant recipients [54]. Use of ROCK inhibitors for essential hypertension in humans is not yet approved but may be a helpful strategy. The effects of fasudil on endothelial function in atherosclerosis is tested in patients with coronary artery disease (CAD). These results suggest that activation of Rho kinase (ROCK) contributes to the reduced NO bioavailability and endothelial dysfunction seen in humans with atherosclerosis [203]. Rho inhibitor, Fasudil, is approved for clinical use in Japan and China for the treatment of cerebral vasospasm, epicardial coronary spasm [58], and microvascular angina [204]. A newer inhibitor, ripasudil, is approved for the treatment of glaucoma [205,206]. The long-term oral treatment with the ROCK inhibitor is effective in improving exercise intolerance in those patients. These results indicate the usefulness of ROCK inhibitors for the treatment of coronary vasospastic disorders [207].

A summary of the clinical trials using ROCK inhibitors for cardiovascular therapy from NIH clinical trials.gov is listed in Table 1. Despite the development of numerous potent ROCK inhibitors by different groups, success in clinical trials is limited due to their global effects on BP reduction and the associated increase in heart rate and decrease in lymphocytes. Recent efforts are focused on the development of ROCK inhibitors for acute applications and for conditions that can be treated by localized drug action, such as in eye drops for the treatment of glaucoma and in inhalers for COPD.

## 5. Conclusions

In conclusion, studies show that RhoGTPases have a pathogenic role in cardiovascular diseases, and inhibition shows promise. However, the physiological role it has in different tissues and multiple regulators that modify its activity warrants a better understanding of stimuli specific, tissue-specific mechanisms. An in-depth review has discussed the limitations of targeting Rho in cardiovascular tissues, and the authors suggest individualized treatments for both upstream activators and downstream mediators of Rho [208]. To accomplish this understanding, the regulatory mechanisms and identifying downstream targets in different tissues are essential.

## Figures and Tables

**Figure 1 cells-08-00551-f001:**
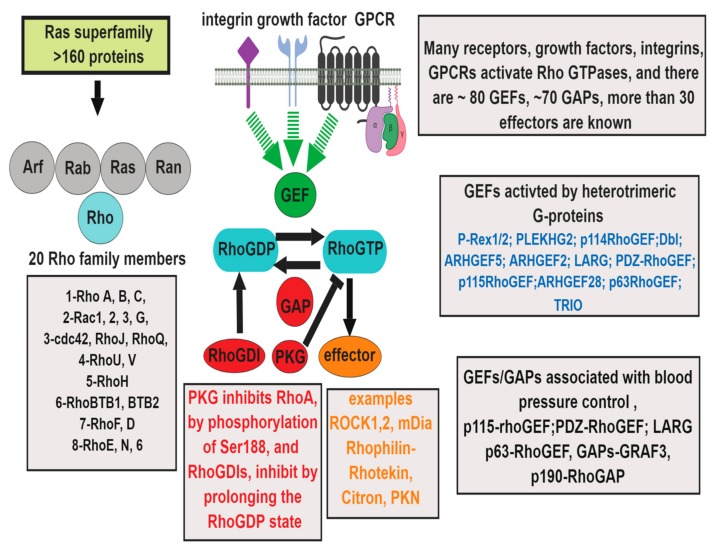
Regulation of Ras-homologous (Rho) GTPase signaling: Rho GTPases belong to the Ras superfamily of more 160 proteins, with six subfamilies [26]. The Rho subfamily has more than 20 members, involved in multiple signal transduction pathways. Control of RhoGTPases is achieved by guanine nucleotide exchange factors (GEFs) and GTPase-activating proteins (GAPs) [6]. Integrins, growth factors, and vasoactive G-protein coupled receptors (GPCRs) linked to heterotrimeric G_12/13_ proteins activate RhoGTPases by mechanisms involving RhoGEFs and RhoGAPs and play an important role in vascular diseases.(Abbreviations: Protein kinase N (PKN), Protein kinase G (PKG), ADP ribosylation factor (Arf ) GTP regulator associated with FAK (Graf), Redox sensing transcriptional repressor (Rex).

**Figure 2 cells-08-00551-f002:**
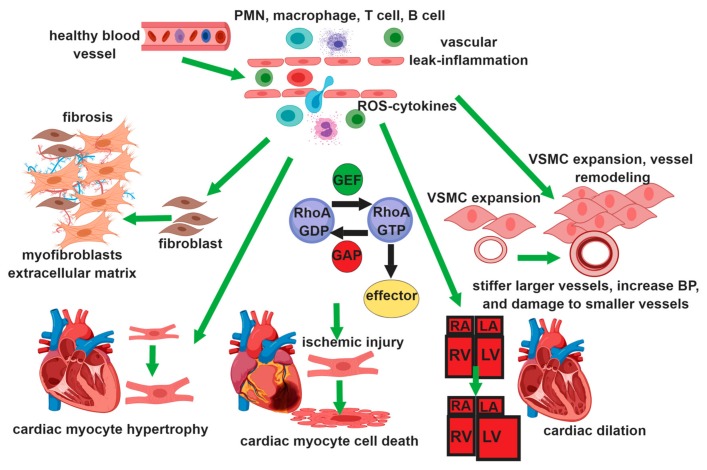
RhoGTPase function in different vascular cells: In the endothelium activation of RhoA, Rac1 control barrier function, leukocyte chemotaxis, ROS production, and inflammation. In smooth muscle cells, RhoA regulates Vascular Smooth Muscle Cell (VSMC) contraction, proliferation, migration, and differentiation. In fibroblasts, RhoA promotes the transformation of fibroblasts to myofibroblasts and increased extracellular matrix production contributing to vascular fibrosis and interstitial fibrosis in the heart and the lung. In the heart, RhoA activity is associated with myocyte hypertrophy, and with apoptosis in ischemic injury.

**Table 1 cells-08-00551-t001:** Clinical Trials Targeting RhoGTPase in Vascular Diseases.

	Trial	Drug
NCT03753269	Early Intracoronary Administration of Fasudil in the Primary PCI of ST-segment-Elevation Myocardial Infarction	Fasudil
NCT03404843	Red Blood Cell ATP Release and Vascular Function in Humans	Fasudil/Saline
NCT00120718	The Effect of Fasudil on Vascular Function in Humans	Fasudil
NCT00670202	Rho Kinase (ROCK) Inhibition in Carotid Atherosclerosis	Fasudil/placebo
NCT03391219	Combined Intravitreal Injection of Bevacizumab and Fasudil Versus Bevacizumab Alone for Macular Edema Secondary to Retinal Vein Occlusion in Previously Treated Patients	Avastin Avastin/fasudil
NCT00498615	A Rho-kinase Inhibitor (Fasudil) in the Treatment of Raynaud’s Phenomenon	Fasudil
NCT00560170	Vascular Effects of Ezetimibe/Simvastatin and Simvastatin on Atherosclerosis	Statin/ROCK biomarker
NCT01823081	Trimetazidine in Pulmonary Artery Hypertension	Trimetazidine/ROCK biomarker
NCT01732718	NCT01732718 Effect of Atorvastatin on Endothelial Dysfunction and Albuminuria in Sickle Cell Disease (ENDO)	Statin ROCK biomarker
NCT02754518	Demonstration of Reverse Remodeling Effects of Entresto (Valsartan/Sacubitril) Using Echocardiography Endocardial Surface Analysis	Entresto ROCK biomarker
NCT00839449	Eicosapentaenoic Acid Cerebral Vasospasm Therapy Study (EVAS)	ROCK biomarker
NCT01065753	Multi-faceted Evaluations Following Weight Reduction in Subjects With Metabolic Syndrome	Rho biomarker
NCT00115830	Rho Kinase in Patients With Atherosclerosis	ROCK biomarker
NCT01069042	Anti-Hypertensive Agent (ACEi) and Heart Function Improvement in Association With Rho Kinase Activity Changes in Human	ROCK biomarker

Additional more specific ROCK inhibitors being tested are 28 Asahi Kasei cerebral vasospasm approved (Japan and China), 165 Kowa glaucoma approved (Japan), diabetic retinopathy phase 2, 45 Kadmon psoriasis phase 2, idiopathic pulmonary fibrosis Phase I.

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
