# Peer review of "RhoGTPase in Vascular Disease"

_cells, 2019, doi:10.3390/cells8060551_

Round 1
Reviewer 1 Report
RhoGTPAse, a critical regulator of vascular system and its function, and is associated with the pathophysiology of multiple cardiovascular diseases. Strassheim et al. have highlighted the importance of RhoGTPAse in the functioning of several important tissues and indicated its role in several cardiovascular diseases in their proposed review. Though the manuscript comprises some critical facts about the functioning of the protein, the review seriously lacks in depth molecular knowledge involved with the functioning of RhoGTPase. Further, the review thinks the manuscript was not constructed with much revisions or editing prior to submission as there exists inconsistent referencing of both style and pattern and understandable codes or numbers in the manuscript.
Author Response
We thank the reviewers for their time for reading and suggestions to improve the scope of the review. We have revised the review substantially to address the reviewers concerns.
Specific comments:- 1. As per this reviewer, the title “RhoGTPase in Vascular Disease” does not appeal to what the reviewer is trying to state. Also, the whole manuscript specifically describes about cardiovascular diseases not other vascular diseases such as peripheral artery disease (PAD), abdominal aortic aneurysm (AAA), carotid artery disease (CAD), arteriovenous malformation (AVM), critical limb ischemia (CLI), pulmonary embolism (blood clots), deep vein thrombosis (DVT), chronic venous insufficiency (CVI).
We acknowledge the reviewer’s concern that the focus was cardiovascular disease and not other vascular diseases. In the revised version we have included the role of RhoGTPase in other vascular disorders such as AVM and aortic aneurysms diseases and other venous diseases in studies where animal models and treatment with ROCK inhibitor was used.
2. Line 42. What does COPD stand for? The authors have not consistently provided the abbreviation throughout the manuscript. Another example is PH. The reviewer strongly suggests that the authors must carefully and consistently construct their abbreviations for the readers.
We apologize for the acronyms, We have corrected this in the revision and expanded the abbreviation when first used in the text.
3. Line 47. The sentence never ends neither does the bracket
We apologize for the omission. We have made corrections
4. Line 49. The authors must be careful of using capital letter such as “Smooth muscles” in this sentence.
We have corrected this.
5. Line 82. The authors must also be careful of using “,” and “.”throughout the manuscript. The Line is not making any sense.
We have rewritten the sentence in new line 86 . The comma before the and was an automatic correction by the text editor.
6. Line 85. “7590342.” What is this supposed to mean?
We apologize for the error in formatting by the reference manager. We have deleted the Pubmed ID number.
7. Line 111. The referencing style changes to “(Lum and Roebuck, 2001, Abid and Razzaque, 2005, Alom-Ruiz and Anilkumar, 2008)” format while throughout it has been numbered. Why?
The error was a conversion of the reference by Endnote. We apologize for missing it. We have corrected the error.
8. Line 147. Why is there a third bracket in the middle of the sentence?
We have deleted the bracket.
9. The figures are not properly described in relevance to the literature described and does indicate any descriptive novelty of their presence in the manuscript.
We have tried to modify the figure to fit the text.
10. As per this reviewer, a number of published reviews have already covered a majority of the literature mentioned in this manuscript. The authors have not justified their intellectual output in their manuscript and have not stated therapeutic options or any current clinical trial currently undergoing to target RhoGTPase.
We agree with the reviewer there are many excellent reviews on the role of Rho in cardiovascular diseases. In our review we have, we have compiled the role of classical RhoGTPases in vascular and inflammatory cell types with an emphasis on their role in disease. We have included a table listing the clinical trials with Rho inhibitors from NIH clinical trials. Gov where either Rho was a target or was a biomarker for treatment.
11. If there are no current trials, they have not priorities the urgency to target the protein.
There is at least six clinical trials listed at NIH clinical trials where Rho is the therapeutic target and eight trials where or it is a biomarker. Also, four newer Rho inhibitors are being tested in Japan for psoriasis, pulmonary fibrosis, glaucoma, and diabetic retinopathy.
12. The reviewer thinks that there lacks an in-depth molecular signaling mechanism associated with each disease involving Rho has not been detailed and should be nailed in more detail.
We have incorporated mechanisms or steps affected by Rho in the diseases described.
Comments and Suggestions for Authors
The article entitled 'RhoGTPase in Vascular Disease' is a timely review that describes the compelling role of RhoA GTPase in particular in various vascular function and diseases. It contributes valuable information but needs very careful revision to make this suitable for readership.
We understand that Rho proteins are important. We have made an effort to parse literature and compilen the latest information available.
Major Points
The scope of the review is not clearly stated. As the authors appreciate that there are at least 23 members of RhoGTPases in the Ras superfamily of monomeric GTPases it is essential to clearly state what is the scope of this review. Is this all the RhoGTPases or RhoA based functions? If it is RhoGTPases in general the authors do not address this comprehensively in the main content. For eg. RhoJ and Rac1 role in various cardiovascular function is completely omitted.
In this review, we have compiled the role of RhoA and Rac1 in vascular diseases. RhoJ the endothelial restricted Rho was shown to be important in tumor biology.
The organisation is poor and detracts the reader from the subject matter. The ROS mediated functional modulation is given as a separate section, which is appropriate. This then becomes useless since the ROS mediated actions are then discussed again in two other major sections (Smooth Muscles and Fibroblasts). All of this needs to be consolidated into one section and moved after Fibroblasts).
We thank the reviewer for the suggestion to improve the organization of the review. We have combined all ROS related sections and moved to one section after fibroblast.
The review discusses the use of Statins which is of great interest to the scientific and clinical Community but lacks a general introduction to why this is the case? This needs to be introduced to the reader at an
appropriate place in the manuscript.
In this review, we have addressed statins only in the context of their effect of inhibition of Rho gtpase. We have mentioned the mechanism by which statins inhibits rho activity.
A list of abbreviation needs to be provided. There is total lack of information on gene names. Using approved gene symbols (based on HUGO nomenclature or ones accepted in the scientific community) is imperative since this will be misleading in future.
The transcriptional control downstream that is regulated by Rho needs to be specified.
What is the specific Rho and Rac we are talking about? eg L179-182 this needs to be clarified throughout the manuscript needs to be tidied to cross check and appreciate the specific cellular functions.
We have included the isoforms of RhoA , Rac1 and Rock where appropriate in the revised version.
Figure 2 and the legend does not make sense complete together. This needs rewriting. The legend does not explain anything including pictorially represented cell/antigen species entering the cell for example.
The manuscript needs to be carefully read through and double checked.
Minor Points
The authors need to very carefully proof read the content for conjunctions, syntax errors, grammatical errors, referencing style and spacing.
Here are a few picked up
The title can be revised since there is discussion of multiple disorders.
As per the suggestion of the first reviewer we have included other vascular diseases where RhoA and Rac are involved.
Figure 1. We need reference for 80 GEFs and 70 GAPs
We have included the reference for this statement in the figure legend and in the text.
78-79. Revise the sentence.
We have revised the sentence
L81- very poor formatting.
We have reformatted the sentence.
L85- what is PH? What is 7590342?
We apologize for the acronym. We have expanded abbreviations in the revised version. The number was PMID for the statement and was not deleted.
L88- revise to’Induces and increase in’ . Ref [26] and [32] IN APPROPRIATE FORMATING
We have revised the sentence.
L89- Cigarette Smoking- Least informative. Vague. Does it belong in this paragraph? A STAND Alone statement with no context
We have rewritten the statement.
L92-93- and 94-95.Does not make sense.
We have revised these lines.
L104-105- Where is the Cysteine in the P loop?
Rho subfamily GTPases that contain a cysteine, Cys18 (Rac1 numbering), located at the end of the P-loop (GXXXXGK(S/T), residues 10 to 17, Rac1 numbering). The Journal of Biological Chemistry280, 31003-31010.
L111- Reference remaining in Harvard Style!!!!
We apologize for the glitch in Endnote. We have corrected the error.
Otherwise too many references missing in that paragraph (L113-122)
We have fixed the references.
L134- Which Rho exactly?
The isoform for that statements was RhoA. We have modified the statement
ROCK is given through out without specifying ROCK1 or ROCK2.
We have specified the isoform where information was available.
L166-Which effector are we talking about?
The effector was ROCK. We have rewritten the sentience.
What is the Smad and which BMP is referred here? (L187). We need to be specific.
It was Smad1 . We have specified the isoform.
L224-232- Why are we discussing this? It need to be either clearly tied into how this related to Rho GTPase function. Otherwise remove.
The section listed preclinical studies with rho inhibitor where therapeutic effects were seen with Rho inhibition.
Submission Date
18 March 2019
Reviewer 2 Report
The article entitled 'RhoGTPase in Vascular Disease' is a timely review that describes the compelling role of RhoA GTPase in particular in various vascular function and diseases. It contributes valuable information but needs very careful revision to make this suitable for readership
Major Points
The scope of the review is not clearly stated. As the authors appreciate that there are atleast 23 members of RhoGTPases in the Ras superfamily of monomeric GTPases it is essential to clearly state what is the scope of this review. Is this all the RhoGTPases or RhoA based functions? If it is RhoGTPases in general the authors do not address this comprehensively in the main content. For eg. RhoJ and Rac1 role in various cardiovascular function is completely omitted.
The organisation is poor and detracts the reader from the subject matter. The ROS mediated functional modulation is given as a separate section, which is appropriate. This then becomes useless since the ROS mediated actions are then discussed again in two other major sections (Smooth Muscles and Fibroblasts). All of this needs to be consolidated into one section and moved after Fibroblasts.
The review discusses the use of Statins which is of great interest to the scientific and clinical Community but lacks a general introduction to why this is the case? This needs to be introduced to the reader at an appropriate place in the manuscript.
A list of abbreviation needs to be provided. There is total lack of information on gene names. Using approved gene symbols (based on HUGO nomenclature or ones accepted in the scientific community) is imperative since this will be misleading in future.
The transcriptional control downstream that is regulated by Rho needs to be specified
What is the specific Rho and Rac we are talking about? eg L179-182 this needs to be clarified throughout the manuscript needs to be tidied to cross check and appreciate the specific cellular functions.
Figure 2 and the legend does not make sense complete together. This needs rewriting. The legend does not explain anything including pictorially represented cell/antigen species entering the cell for example.
The manuscript needs to be carefully read through and double checked.
Minor Points
The authors need to very carefully proof read the content for conjunctions, syntax errors, grammatical errors, referencing style and spacing.
Here are a few picked up
The title can be revised since there is discussion of multiple disorders
Figure 1. We need reference for 80 GEFs and 70 GAPs
L78-79. Revise the sentence.
L81- very poor formatting.
L85- what is PH? What is 7590342?
L88- revise to’Induces and increase in’ . Ref [26] and [32] IN APPROPRIATE FORMATING
L89- Cigarette Smoking- Least informative. Vague. Does it belong in this paragraph? A STAND Alone statement with no context
L92-93- and 94-95.Does not make sense.
L104-105- Where is the Cysteine in the Ploop?
L111- Reference remaining in Harvard Style!!!!
Otherwise too many references missing in that paragraph (L113-122)
L134- Which Rho exactly?
ROCK is given through out without specifying ROCK1 or ROCK2
L166-Which effector are we talking about?
What is the Smad and which BMP is referred here? (L187). We need to be specific.
L224-232- Why are we discussing this? It need to be either clearly tied into how this related to Rho GTPase function. Otherwise remove.
Author Response

(The authors gave the same response as above.)

Round 2
Reviewer 1 Report
Dear Authors,
The review reads much better compared to the last version. The reviewer is happy with the current changes. Minor comments:-
Kindly change the Line 43-50 from bold to normal.
The paragraphing space between paragraphs on page 9 is different compared to the rest. Please kindly validate the same.
Author Response
We thank the reviewer for his time and suggestions to make the review better.
We have unbolded the lines from 41-50.
We have corrected the paragraph spacing on page 9
Reviewer 2 Report
The revision of the content is satisfactory. Full form of gene names, cofactors need to be given throughout.
Author Response
We thank the reviewers for the time and suggestions to make the review better.
We have expanded the names of the genes as suggested by the reviewer.